# In Vitro Characterization of Renal Drug Transporter Activity in Kidney Cancer

**DOI:** 10.3390/ijms231710177

**Published:** 2022-09-05

**Authors:** Pedro Caetano-Pinto, Nathanil Justian, Maria Dib, Jana Fischer, Maryna Somova, Martin Burchardt, Ingmar Wolff

**Affiliations:** Department of Urology, University Medicine Greifswald, 17475 Greifswald, Germany

**Keywords:** drug transporter, renal cell carcinoma, proximal tubule epithelial cells, nephrotoxicity

## Abstract

The activity of drug transporters is central to the secretory function of the kidneys and a defining feature of renal proximal tubule epithelial cells (RPTECs). The expression, regulation, and function of these membrane-bound proteins is well understood under normal renal physiological conditions. However, the impact of drug transporters on the pathophysiology of kidney cancer is still elusive. In the present study, we employed different renal cell carcinoma (RCC) cell lines and a prototypical non-malignant RPTEC cell line to characterize the activity, expression, and potential regulatory mechanisms of relevant renal drug transporters in RCC in vitro. An analysis of the uptake and efflux activity, the expression of drug transporters, and the evaluation of cisplatin cytotoxicity under the effects of methylation or epidermal growth factor receptor (EGFR) inhibition showed that the RCC cells retained substantial drug transport activity. In RCC cells, P-glycoprotein was localized in the nucleus and its pharmacological inhibition enhanced cisplatin toxicity in non-malignant RPTECs. On the other hand, methylation inhibition enhanced cisplatin toxicity by upregulating the organic cation uptake activity in RCC cells. Differential effects of methylation and EGFR were observed in transporter expression, showing regulatory heterogeneity in these cells. Interestingly, the non-malignant RPTEC cell line that was used lacked the machinery responsible for organic cation transport, which reiterates the functional losses that renal cells undergo in vitro.

## 1. Introduction

The excretion of metabolites and xenobiotics is a paramount physiological function of the kidneys. In addition to filtration at the glomerular level, the renal proximal tubules are responsible for active drug transport, which can move small molecules against steep concentration gradients from systemic circulation into the pre-urine [1]. This process is mediated by an array of different membrane carriers belonging to the solute carrier (SLC) and ATP-binding cassette (ABC) families. The expression of drug transporters greatly impacts the physiology of renal proximal tubule epithelial cells (RPTECs) [2]. Highly polarized in nature, RPTECs enable specific transporters to be exclusively localized in the basolateral or apical membranes of the cells, functioning in tandem. The activity of drug transporters can indirectly regulate aspects of cellular homeostasis, such as ATP production, given the high energy demands of ABC-family carriers [3]. Nephrotoxicity is a phenomenon directly related to drug transport activity. Thanks to the blood-filtering functions of RPTECs, cytotoxic drugs transit through these cells via active transport. These drugs can accumulate in RPTECs depending on the doses administered and on the activity of both the uptake and efflux membrane transporters [4]. In particular, molecules with low permeability require active transport to be extruded from the cells, and their retention can lead to the loss of the renal epithelium.

In the kidney, about thirty transporters belonging to the SLC and ABC families have been identified [5]. In RPTECs, to date, eleven of these transporters have been recognized as pharmacokinetic and nephrotoxicity regulators, and their functional testing is required by the regulatory agencies during the pre-clinical development of new drugs [6]. The expression profile of drug transporters and their activity is a hallmark feature of the RPTEC phenotype and a distinctive physiological characteristic [1]. While the role of these proteins is well understood in RPTECs under normal physiological conditions, their impact on the pathophysiology of renal diseases is far less clear [7,8], in particular in kidney cancers.

The most common type of kidney cancer is renal cell carcinoma (RCC), which represents the third most frequently diagnosed type of urologic tumor and is among the ten most commonly diagnosed tumor types overall [9]. RCC originates from RPTECs following a differentiation process of these cells that ultimately culminates in the acquisition of a malignant phenotype [3]. A key feature of RCC is the loss of the Von Hippel–Lindau (VHL) protein, an E3 ligase that targets proteins for proteasomal degradation [10]. Importantly, the VHL protein mediates the degradation of hypoxia-inducible factors (HIFs). HIFs mediate the response to variations in cellular oxygen levels. In the absence of the VHL protein, the deregulated HIF activity pushes the cellular metabolism towards a predominant glycolytic state [11,12]. This metabolic shift is central to RCC physiology and is responsible for maintaining the tumor phenotype, including its angiogenic and inflammatory nature [13]. By releasing vascular endothelial factors (VEGFs), tumors recruit the renal vasculature to increase their blood supply and meet their energetic needs [14]. Extensive cytokine secretion generates an inflammatory firewall that safeguards RCC from the activity of immune cells (e.g., leukocytes and lymphocytes), therefore facilitating tumor growth [3].

RCC pathophysiology is well known to be associated with two different cellular regulatory processes: DNA methylation [10] and epidermal growth factor (EGFR) signaling [15,16]. Under normal physiological conditions, methylation can function as a gene activator during development, regulating DNA transcription. In RCC, the proportion of methylation islands in the DNA is substantially increased relative to that of benign tissue. Hypermethylation is associated with the loss of the VHL protein [17] and the deregulation of genes encoding the enzymes and carrier proteins responsible for drug metabolism and disposition, including drug transporters [18]. EGFR tyrosine kinase-mediated signaling pathways are central to the regulation of the activity and function of epithelial cells, as they govern growth, differentiation, cellular permeability, and adhesion among many other processes [19,20,21]. The defective, overexpressed, or constitutive activation of EGFR is associated with the proliferation of several types of epithelial cancers [22]. Enhanced hypermethylation and EGFR expression are associated with advanced and metastatic RCC stages [18].

In the present study, we investigate the drug transport activity, expression, and nephrotoxic responses of the established RCC cell lines CAKI-1 [23] and 786-O [24] relative to the prototypical RPTEC-TERT1 cell line [25], an immortalized RPTEC-derived cell line representative of a healthy proximal tubule epithelium. Drug resistance is a well-documented phenomenon in cancer, where the membrane secretory machinery assumes a protective function, preventing the accumulation of compounds detrimental to cell survival [26]. Understanding the functional changes that RPTECs undergo during their transition into carcinoma cells is important to elucidate the role that drug transporters play in RCC physiology, namely cell survival, considering that their conventional secretory function is absent in cancer cells. Beyond drug resistance, membrane transporters can indirectly regulate metabolic activity by controlling the intracellular disposition of metabolic co-factors and bi-products [27,28]. From a clinical perspective, the expression of drug transporters may reflect resistance against systemic therapy in RCC and may be potentially employed as a prognostic or predictive tool [29,30]. Our main objective was to revisit RCC drug transport activity in vitro by employing specific fluorescent substrates, and characterize the impact of methylation and EGFR inhibition on the expression of the characteristic renal drug transporters, namely organic cation transporters 1 and 2 (OCT1 and OCT2), P-glycoprotein (P-gp), and the breast cancer resistance protein (BCRP). Together, these uptake and efflux carriers are responsible for handling the majority of compounds known to be renally secreted [31,32,33]. Moreover, this study explored the functional differences and similarities of both malignant and non-malignant RPTEC models in conventional cell culture.

## 2. Results

### 2.1. Drug Transport Activity in Non-Tumor and RCC Cell Lines

All cell lines analyzed showed substantial efflux activity, but a limited organic cation uptake function. The results of a non-linear regression analysis to derive the apparent K_m_ and V_max_ parameters are listed in Table 1.

Interestingly, non-tumor RPTEC-TERT1 and tumor 786-O cells showed no dose-dependent inhibition in ASP^+^ uptake, indicating limited organic anion uptake activity. On the other hand, the tumor line CAKI-1 showed inhibitable and dose-dependent ASP^+^ uptake (Figure 1). There were no major differences observed in the CAKI-1 ASP^+^ K_m_ values between the uninhibited and inhibited uptake (within 2-fold; Table 1), which is an indication that the uptake inhibition by imipramine did not affect the affinity of the organic cation transporters for the substrate. These findings show that uptake activity can significantly differ between RCC cell models, and that RPTEC-TERT1, a supposed cell line representative of a healthy renal epithelium, lacks a key functionality of proximal tubules.

Calcein accumulation was evident in all cell lines (Figure 2). The efflux inhibition yielded significant retention of the substrate, which represents a clear indication of P-gp activity. The K_m_ values between the uninhibited and inhibited efflux in all cell lines were below 2-fold (Table 1), indicating that the P-gp inhibition by valspodar did not change the affinity of the transporter for calcein.

BCRP activity was also evident, with all cell lines displaying dose-dependent Hoechst33342 uptake (Figure 3). The efflux inhibition by KO143 was observed; however, it was observed to a lesser extent relative to calcein, showing that BCRP transport activity is less predominant in these cells. The 786-O cells showed the highest Hoechst33342 accumulation upon inhibition. The K_m_ values between the uninhibited and inhibited efflux were below 2-fold in all cell lines and within a similar range across all cell lines (Table 1), indicating that these cells shared similar BCRP transport activity.

### 2.2. Expression of Renal Drug Transporter upon Methylation or EGFR Inhibition

The uptake and efflux of renal drug transporters were analyzed to determine differences in their gene expression profiles after treatment with either the methylation inhibitor decitabine or the EGFR inhibitor cetuximab (CTX). The relative expression across all cell lines is listed in Table 2.

Overall, our results show that, in terms of absolute gene expression, P-gp was predominantly expressed in all cell lines, with higher BCRP expression in the tumor cells compared to the non-tumor cell line (RPTEC-TERT1). Remarkably, no cell lines expressed OCT2, and the expression of OCT1 was limited in all cell lines (Figure 4). In the CAKI-1 cells, the decitabine treatment recovered OCT2, albeit to a limited extent. The OCT1 expression was upregulated by methylation inhibition in the RPTEC-TERT1 cells. The P-gp expression was upregulated in the RPTEC-TERT1 and CAKI-1 cells in the presence of CTX. The BCRP expression was downregulated by CTX in the CAKI-1 cells and, conversely, was upregulated in the 786-O cells.

Moreover, the efflux transporters MRP4 and MRP2 were highly expressed in the tumor cells (Figure 5). In the RPTEC-TERT1 cells, the MRP4 expression was elevated and the MRP2 levels were limited. The MATE1 and MATE2 transporters were absent in the RPTEC-TERT1 cells. MATE1 was also absent from the 786-O cells, and its expression was increased by CTX in the CAKI-1 cells. The MATE2 expression was downregulated by CTX in the CAKI-1 cells and, conversely, was upregulated in the 786-O cells. The uptake CTR1 transporter was also highly expressed across all cell lines, and in the CAKI-1 cells, its levels increased after both the decitabine and CTX treatment.

These results show that tumor cells—in particular, CAKI-1 cells—respond to CTX treatment and decitabine exposure by altering the expression profile of drug transporters. The 786-O cells were sensitive to methylation inhibition. Interestingly, the RPTEC-TERT1 cells were shown to be associated with a lower expression of key renal drug transporters, as opposed to the tumor cell lines analyzed.

### 2.3. Cellular Localization of BCRP and OCT1

Immunofluorescent staining of BCRP and OCT1 revealed that in all the cell lines analyzed, the expression of BCRP and OCT1 was seemingly dispersed throughout the intracellular space and cellular boundaries in endosomes. Both transporters were not predominantly localized in the cell membrane (Figure 6).

### 2.4. Cellular Localization of P-Glycoprotein in Non-Tumor and RCC Cell Lines

Immunofluorescent staining of P-gp revealed the cellular localization of the transporter in tumor and non-tumor renal epithelial cells (Figure 7).

The results showed that in the RPTEC-TERT1 cells, P-gp was seemingly dispersed in the cytoplasm and the nuclear region. In the CAKI-1 and 786-O cells, P-gp was confined to the nucleus of the cells. Under normal physiological conditions, P-gp is expressed in the apical side of the plasma membrane of renal proximal tubule cells. These findings indicate that, for both the non-tumor and tumor cell lines, P-gp localization significantly diverged from native physiology. Moreover, the phalloidin staining revealed that the RPTEC-TERT1 cells maintained a more characteristic epithelial morphology relative to the tumor cell lines. F-actin filaments were predominantly concentrated in the cellular boundaries and the cells grew into a regular monolayer.

### 2.5. Effects of Methylation or EGFR Inhibition on Cisplatin Cytotoxicity

Cisplatin cytotoxicity was evaluated after the cells were exposed to either decitabine or CTX. This highly nephrotoxic drug requires active transport via organic cation transporters to accumulate intracellularly.

In the RPTEC-TERT1 cells, cisplatin reduced cell viability to about 40% relative to untreated cells. The CTX pre-treatment did not affect the toxicity (Table 3). Methylation inhibition with decitabine resulted in a 2.8-fold reduction in the half-maximal effective concentration (EC50). The maximal toxicity of 40% was not affected by decitabine (Figure 8).

In the CAKI-1 cells, cisplatin toxicity had a pronounced effect, with the viability reduced to about 10% with the maximal cisplatin concentration used (1000 µM). The CTX pre-treatment did not affect toxicity. Methylation inhibition with decitabine resulted in a 2-fold reduction in the EC50. The maximal toxicity observed was not affected by decitabine.

In the 786-O cells, cisplatin also reduced cell viability to about 40% relative to untreated cells. In these cells, decitabine and CTX pre-treatments did not seem to affect toxicity.

### 2.6. Changes in Uptake and Efflux Activity after Methylation or EGFR Inhibition

The uninhibited uptake and efflux of fluorescent substrates were used to assess changes in drug transport activity promoted by exposure to decitabine or CTX. Although minor, the changes in transport activity were determined after ASP^+^ and Hoechst33342 incubation (Figure 9).

ASP^+^ accumulation increased in the RPTEC-TERT1 and CAKI-1 cells after the decitabine pre-treatment, which indicated an increase in OCT activity. CTX seemingly did not affect ASP^+^ uptake. Hoechst33342 accumulation decreased in the RPTEC-TERT1 cells after CTX exposure, indicating an upregulation in the BCRP efflux activity, which resulted in reduced intracellular substrate retention. In the CAKI-1 cells, CTX increased the accumulation of Hoechst33342, indicating the downregulation of BCRP activity. Decitabine exposure seemingly did not affect the Hoechst33342 retention, and in the 786-O cells, the retention was not affected by either methylation or EGFR inhibition.

### 2.7. Effects of P-Glycoprotein Inhibition on Cisplatin Cytotoxicity

Cisplatin effects were evaluated in the presence of valspodar to determine the impact of P-gp inhibition on cytotoxicity. Cisplatin is not a P-gp substrate; therefore, its role in cisplatin toxicity is non-drug-transport-mediated.

In tumor cells, the addition of valspodar to cisplatin did not affect EC50 or the toxicity observed with the maximal cisplatin concentration, for which a viability of 10% and 40% in the CAKI-1 and in 786-O cells was found, respectively (Figure 10). On the other hand, in the RPTEC-TERT1 cells, the co-incubation of valspodar and cisplatin increased EC_50_ by 1.4-fold (Figure 10). The maximal cisplatin toxicity in the presence or absence of the inhibitor was about 20%. These results showed that P-gp inhibition differentially affects cytotoxicity in tumor and non-tumor renal cells.

Exposure to 200 µM cisplatin, a concentration around the EC_50_ values determined, was used to stress the cells while not inducing extensive cell death. Under these conditions, the onset of apoptosis was evaluated using fluorescently labeled annexin V. Annexin V binds to phosphatidylserine, a phospholipid that is translocated to the outer cell membrane during the onset of apoptosis [34]. In tumor cells, no substantial differences were observed in the annexin V stains in the presence or absence of P-gp inhibition. In the RPTEC-TERT1 cells, the annexin V stain was more pronounced, increasing after cisplatin exposure relative to untreated cells and decreasing in the presence of valspodar, indicating that P-gp inhibition negatively impacts the apoptotic response in these cells, although slightly.

## 3. Discussion

Drug resistance is a well-documented phenomenon in many cancer types. It can be inherent to the malignant cells or acquired as the result of tumor progression. Tumor cells resist the effects of therapeutical agents by preventing their accumulation, hence reducing exposure [35]. The expression and activity of membrane drug transporters is at the heart of this process, and they can act to preclude the uptake of drugs and enhance their efflux [36,37]. Cells can also rewrite their regulatory pathways and render targeted therapies ineffective. In this process, tumors alter their physiology in ways that ensure their survival is not affected by a previously effective drug [38]. Tumor heterogeneity and phenotypical selection are, arguably, major factors contributing to drug resistance. Exposure to a particular drug may inadvertently act as a selection agent that eradicates a sensitive cell population and enables a resistant subpopulation to thrive [39]. These are important considerations when employing in vitro tools to study the activity of cancer cells. Cell lines represent the physiology of a single tumor and have been further isolated and maintained in culture.

Functionally, the CAKI-1 cells were the only cells that displayed organic cation uptake activity, which was evident from the inhibitable accumulation of ASP^+^. The lack of this activity in the other cell lines was due to the absence of OCT2 expression in both the RCC cells and the RPTEC-TERT1 cells, and limited OCT1 expression in all cell lines. This observation was compounded by the fact that primary human RPTECs (HRPTECs) expressed OCT2 and showed substantial ASP^+^ accumulation and uptake inhibition (Appendix C and Appendix D). Methylation inhibition in the CAKI-1 cells recovered the OCT2 expression, albeit to a limited extent, which is an effect that was not observed in the other cell lines. The recovery of the OCT2 expression contributed to enhanced cisplatin toxicity and ASP^+^ uptake in the CAKI-1 cells after the decitabine treatment. These findings are consistent with previous reports showing that methylation inhibition improves the toxicity of platinum-based chemotherapeutics in RCC by recovering the activity of organic cation transporters [18,29]. Interestingly, in the RPTEC-TERT1 cells, the decitabine treatment enhanced cisplatin toxicity and ASP^+^ uptake, while upregulating the OCT1 expression (Figure 4). Cisplatin is reported to be a substrate for copper transporter 1 (CTR1) [40,41], which was found to be relatively abundant in all cell lines (Figure 5). The expression of CTR1 in the tested cell lines was seemingly not a determining factor for cisplatin toxicity, given the different cytotoxic responses to cisplatin observed, albeit similar CTR1 expressions. Moreover, CAKI-1 was the only cell line other than HRPTEC that retained the expression of MATE1 and MATE2, the efflux transporters that physiologically complement OCT2 activity [42,43]. Interestingly, and given the fact that MATE activity is driven by an electrochemical gradient [44], incubating CAKI-1 cells in acidic conditions (pH = 6) reduced cisplatin toxicity and decreased ASP^+^ uptake (Appendix E Figure A3). MATE transporters could play a more determinant role in cisplatin toxicity in the absence of OCT2, and further investigations into this topic are of interest in this research line.

Overall, in the CAKI-1 cells, the OCT2 expression could be regulated by methylation, while the expression of MATE1 and MATE2 is seemingly regulated by EGFR, which was evident by the deregulated expression following CTX treatment. The 786-O cells expressed MATE1, but lacked MATE2, and these transporters were both absent in the RPTEC-TERT1 cells (Figure 5). These cells showed decreased cisplatin toxicity relative to the CAKI-1 cells. These findings highlight the heterogeneity of RCC cell lines and the stark functional differences in organic cation transport activity. Furthermore, the regulatory networks behind the expression of drug transporters were different between the cell lines, as the CAKI-1 cells were sensitive to methylation and EGFR inhibition to a higher extent than the 786-O cells. On the other hand, the RPTEC-TERT1 cells seemingly lacked organic cation uptake and efflux machinery, except for OCT1. This fact can be considered a significant downside for a model representing human RPTECs.

The evident accumulation of calcein in all cell lines illustrated that P-gp represents a major player in efflux activity, in both RCC and non-malignant cells. Calcein is also a substrate for multidrug resistance transporters (MRPs), although with a lower affinity than P-gp, and the expression of MRP2 and MRP4 in the RCC cell lines could complement the activity of P-gp [45]. In the RPTEC-TERT1, CAKI-1, and HRPTECs, the P-gp expression was upregulated after treatment with CTX, which indicated that the upstream regulatory pathways governed by EGFR can dictate P-gp expression in both RCC and healthy renal RPTECs. However, at a functional level, CTX did not impact the calcein accumulation. The Hoechst33342 retention revealed that BCRP was also an active efflux transporter in the cell lines analyzed, albeit to a lesser extent than P-gp, which was evident from the reduced accumulation upon selective inhibition. Interestingly, CTX promoted a marked reduction in BCRP expression in the CAKI-1 cells, and a slight reduction in the Hoechst33342 accumulation, indicating a functional downregulation. These observations show that the mechanism behind BCRP regulation differs between non-malignant and tumor cell lines and among tumor cell lines themselves, given that in the 786-O cells, decitabine yielded a slight upregulation in the BCRP gene expression.

The expression of drug transporters at the gene level does not dictate function, which is further dependent on the protein expression and crucially on the adequate localization of the transporters in the cell membrane. Immunofluorescent characterization showed that in the RPTEC-TERT1, CAKI-1, and 786-O cells, BCRP and OCT1 were dispersed in the cytoplasm and in the vicinity of the nucleus. Drug transporters are trafficked to the membrane and can reside in endosomes. Our findings show that, although expressed at the gene level, these transporters were not predominantly localized in the cell membrane (Figure 6). Our activity results reflect this fact, namely regarding BCRP, which was expressed and active, although limited accumulation was determined upon inhibition. The precluded localization of membrane-bound proteins is derived from a poor polarization of the cell in conventional culture [46]. Cells lack defined basolateral and apical membranes, which in turn results in the abhorrent localization of drug transporters. Arguably, the cells used in our study could gain functional activity in culture conditions that facilitate a better polarization. Noteworthy was the fact that P-gp was predominantly dispersed in the cytoplasm, but only in the RPTEC-TERT1 cells (Figure 7A). In the CAKI-1 and 786-O cells, P-gp was localized in the nucleus (Figure 7B,C), which is potentially a distinctive feature of RCC cells. The nuclear expression of P-gp has been reported in different cancer cell lines [32,47] and is believed to be associated with the enhanced protection of genetic information from chemical-induced damage, which is a drug-resistance mechanism. The calcein accumulation observed in the RCC cell lines was therefore driven by both cytoplasmic and nuclear accumulation. Moreover, the sensitivity of the RPTEC-TERT1 cells to cisplatin seemingly increased after P-gp inhibition (Figure 10), an effect that was not observed in the RCC cells. P-gp activity has been previously associated with the regulation of cell death in cancer, in a drug transport-independent mechanism [48]. In leukemic cells, P-gp inhibition leads to an increase in apoptosis activity [49]. Our results show that P-gp inhibition reduced the apoptotic activity in the RPTEC-TERT1 cells, indicating that P-gp plays a role in regulating cell death. Arguably, the nuclear P-gp expression in RCC cells could fulfill an apoptosis mitigation role that contributes to tumor cell survival and drug resistance.

The expression of organic anion transporters 1 and 3 (OAT1 and OAT3) was also analyzed in our study. However, no cell line, including HRPTEC, expressed these key renal transporters. These findings are unsurprising, since OAT loss in isolated renal cells is a well-known phenomenon [50]. Therefore, we were unable to ascertain if the RCC cells lost OAT expression due to their transformed phenotype or if this was a result of the culture conditions, as is believed to be the case for the RPTEC-TERT1 cells and the HRPTECs.

In the present study, we determined that the RCC cell lines CAKI-1 and 786-O displayed differential expression of renal drug transporters as well as uptake and efflux activity. Methylation and EGFR inhibition also yielded different effects in transporter expression, highlighting the regulatory disparity between cell lines representative of kidney cancer. The RPTEC-TERT1 cells were shown to lack expression and activity of important renal transporters compared to primary RPTECs. This loss of activity is known to occur when RPTECs are isolated and kept in culture [50]. The fact that the RPTEC-TERT1 cells were artificially immortalized may have contributed to their resemblance to RCC cell lines, which are immortalized as well due to their malignant phenotype. Nonetheless, RPTEC-TERT1 cells do maintain key RPTEC traits, e.g., their epithelial morphology and P-gp expression. The RCC cell lines used in this study also showed enhanced EGFR expression relative to the expression control (Appendix G Figure A5) when compared to RPTEC-TERT1 cells and HRPTECs, which is a well-documented characteristic of RCC tumors [15,16]. Overall, our results show that non-malignant and tumor RCC cell lines share functional similarities, and that allegedly, no cell line alone is truly representative of either the RPTEC or the RCC phenotype in vitro. Interestingly, the CAKI-1 cells maintained substantial drug transport activity with recoverable OCT2 expression, making these cells a relevant model for studying organic cation activity and regulation.

## 4. Materials and Methods

### 4.1. Cell Culture

Human primary renal proximal tubule epithelial cells (HRPTECs; Biopredic, Saint Grégoire, France) and immortalized renal proximal tubule epithelial cells (RPTEC-TERT1; LGC/ATCC, Teddington, UK) were used as non-malignant (healthy) models. CAKI-1 (LGC/ATCC, Teddington, UK) and 786-O cells were used as RCC models. All chemicals and reagents used in this study are listed in Appendix B Table A3. HRPTECs and RPTEC-TERT1 cells were cultured using DMEM F-12 high glucose media (Gibco, Carlsbad, CA, USA) supplemented with ITS (Lonza, Basel, Switerzerland) (10 μg/mL of insulin, 5.5 μg/mL of transferrin, and 5 ng/mL of sodium selenite), 10 ng/mL of epidermal growth factor (Merck, Darmstadt, Germany), 36 ng/mL of hydrocortisone, 100 U/mL of penicillin, 100 μg/mL of streptomycin (PAN biotech, Aidenbach, Germany) (5% *v*/*v*), and either 1% (*v*/*v*) FCS (PAN biotech) in HRPTECs or 10% (*v*/*v*) FCS in the RPTEC-TERT1 cells. CAKI-1 cells were cultured in Eagle MEM (PAN biotech) supplemented with 1 mM sodium pyruvate (PAN biotech), 100 U/mL of penicillin, 100 μg/mL of streptomycin (5% *v*/*v*), and 10% (*v*/*v*) FCS. The 786-O cells were cultured in RPMI 1640 (PAN biotech) supplemented with 100 U/mL of penicillin, 100 μg/mL of streptomycin (5% *v*/*v*), and 10% (*v*/*v*) FCS. The HRPTECs and RPTEC-TERT1 cells were cultured in microplates at a density of 2,500,000 cells/mL while the CAKI-1 and 786-O cells were grown at an initial density of 100,000 cells/mL. All assays were performed after the cells reached approximately 95% confluency, usually within 5 days for the HRPTECs and RPTEC-TERT1 cells and 2 days for the CAKI-1 and 786-O cells. CTX and cisplatin were acquired from the University Medicine Greifswald’s central pharmacy.

### 4.2. Gene Expression

The expression of specific renal membrane drug transporters was determined using the TaqMan gene expression system (ThermoFisher, Waltham, MA, USA). Cells were cultured in 6-well plates until 95% confluency prior to treatment with 100 nM Aza or 100 µg/mL of CTX for 24 h; untreated cells were used as controls. The total RNA was extracted using the RNAeasy kit (Qiagen, Venlo, The Netherlands) and cDNA was generated using a high-capacity reverse transcriptase kit (Applied Biosystems, Waltham, MA, USA) according to manufacturer specifications. The gene probes used are listed in Appendix A.

### 4.3. Functional Assays

Drug transport activity was evaluated by determining the intracellular accumulation of fluorescent substrates in the presence or absence of specific inhibitors [19,45]. The cationic substrate 4-[4-(dimethylamino) styryl]-1-methyl pyridinium iodide (Merck) (ASP+; excitation: 475 nm; emission: 609 nm) together with imipramine (Merck) (50 µM), a potent organic cation transporter inhibitor, were used to determine uptake activity. ASP+ requires active transport to accumulate intracellularly, since it does not passively diffuse through the plasma membrane. To evaluate efflux activity, Hoechst33342 (Merck) (excitation: 350 nm; emission: 460 nm) and calcein-AM (Merck) (excitation: 488 nm; emission: 520 nm) were used as prototypical substrates for BCRP and P-gp, respectively. These substrates passively diffuse into cells and are secreted via active transport. Hoechst33342 binds DNA and its intracellular retention was evaluated using BCRP inhibitor KO143 (Merck) (25 µM). Calcein-AM is metabolized into its fluorescent form—calcein—by esterase activity in the cytoplasm. Calcein fluorescence indicated that calcein-AM was not being secreted via P-gp, and its accumulation was evaluated using the specific P-gp inhibitor valspodar (Merck) (25 µM). Function and activity assays were performed at 37 °C using Krebs buffer. Microplates were incubated for 15 min (ASP+), 30 min (Hoechst33342), or 45 min (calcein-AM). Subsequently, the microplates were washed twice with cold buffer, and the cells incubated with ASP+ and calcein-AM were lysed with 0.1% TritonX-100 (Merck) in Krebs buffer (Merck) to disperse the fluorescent signal in suspension in order to facilitate measurement. Fluorescent intensities were acquired using a Tecan Infinity 2000 microplate reader. The effects of CTX (100 µg/mL) and decitabine (100 µM) on the uptake and efflux activity of the cells were evaluated by accessing the retention of specific fluorescent substrates, without inhibition, after the cells were pre-treated with CTX or decitabine for 24 h.

### 4.4. Toxicity Assays

Cells were pre-treated with 100 µg/mL of CTX or 100 mM decitabine CTX in culture media for 24 h before cisplatin exposure. Cisplatin was prepared at the maximum concentration of 1000 µM and a step dilution of 1:4. Cells were exposed to the drug for 6 h and afterwards washed and cultured further in a drug-free culture medium for an additional 24 h. Following the recovery period, cells were washed and incubated with Presto Blue reagent (Invitrogen, Waltham, MA, USA) prepared at a dilution of 1:20 in a culture medium for 1.5 h. Immediately after incubation with Presto Blue (Invitrogen, Carlsbad, CA, USA), the fluorescence intensity was acquired using a Tecan Infinity 2000 microplate reader (Männedorf, Switzerland) at an excitation of 530 nm and a 590 nm emission.

An annexin V-FITC assay kit (PromoCell, Heidelberg, Germany) was used to determine apoptosis induction. Cells were cultured in Cell Carrier Ultra microplates (Perkin Elmer, Waltham, MA, USA) and exposed to 200 µM cisplatin for 6 h, followed by a 24 h recovery period, a subsequent wash with HBSS, and incubation with Annexin V-FITC (dilution: 1:20) in binding buffer. Subsequently, cells were washed with HBSS, fixed with 2% paraformaldehyde solution, and stained with Hoechst33342 (1:1000) to highlight the nucleus. Annexin V-FITC fluorescent intensity was acquired using a Tecan Infinity 2000 microplate reader (excitation: 490 nm; emission: 520 nm). Images were acquired using a Keyence BZ9000 fluorescent microscope (Osaka, Japan).

### 4.5. Immunofluorescent Characterization

Cells were cultured in an 8-well slide chamber (LabTek/Nunc, Roskilde, Denmark) until confluency and subsequently washed twice with HBSS before fixation with 2% paraformaldehyde solution. The antibody conditions used are listed in Appendix B. Initially, an HBSS solution with 0.1% Triton-X was used to permeabilize the cells for a minimum of 30 min. Primary antibodies were incubated overnight at 4 °C in a solution of 0.1% Triton-X and 1% BSA (*v*/*v*) in HBSS. The following day, samples were washed for a minimum of 10 min three times (0.1% Triton-X and 1% BSA in HBSS). Secondary antibodies were incubated for two hours at room temperature and subsequently washed for a minimum of 10 min three times. Slides were mounted and imaged using a Zeiss Axio Observer Z1 fluorescent microscope (Zeiss, Jena, Germany).

### 4.6. Data Analysis

All data were analyzed using GraphPad Prism 8 (La Jolla, CA, USA). Fluorescent functional results were fitted using Michaelis–Menten kinetics to derive the apparent K_m_ and V_max_ drug transport activity parameters. The qPCR results were analyzed according to the 2-ΔΔCt method [51]. Statistically significant differences in the expression between untreated and CTX pre-treated cells were estimated using a 2-way ANOVA. The maximal effective concentration (EC50) was determined using a non-linear analysis of the dose-response cytotoxic effects after cisplatin treatment and recovery. Immunofluorescent images were processed using the open-source software Fiji [52].

## 5. Conclusions

Our results show that RCC cell lines and non-tumor cells retain substantial drug transport activity, namely in terms of efflux. Methylation and EGFR inhibition showed substantial regulatory heterogeneity between the cells analyzed. The RPTEC-TERT1 non-malignant renal cells lacked key organic cation transport machinery, which represents a limitation for its use as a renal functional model.

## Figures and Tables

**Figure 1 ijms-23-10177-f001:**
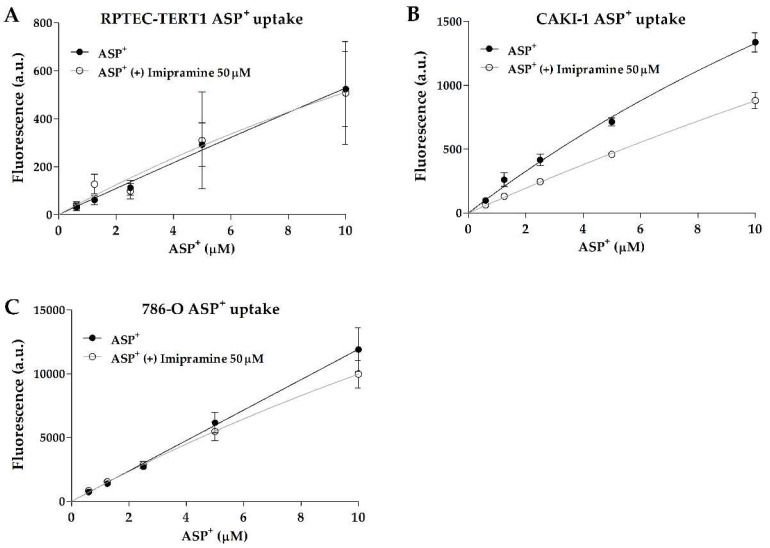
**Organic cation transport activity.** Uptake of ASP^+^ in the presence or absence of OCT inhibitor imipramine. Functional data derived from fluorescent assays were fitted according to Michaelis–Menten kinetics to evaluate the drug transport activity in RPTEC-TERT1 cells (**A**), CAKI-1 cells (**B**), and 786-O cells (**C**). The reduced uptake of ASP^+^ after inhibition indicated the presence of active organic cation uptake transport. Functional data were obtained from a minimum of two independent assays, performed in triplicate.

**Figure 2 ijms-23-10177-f002:**
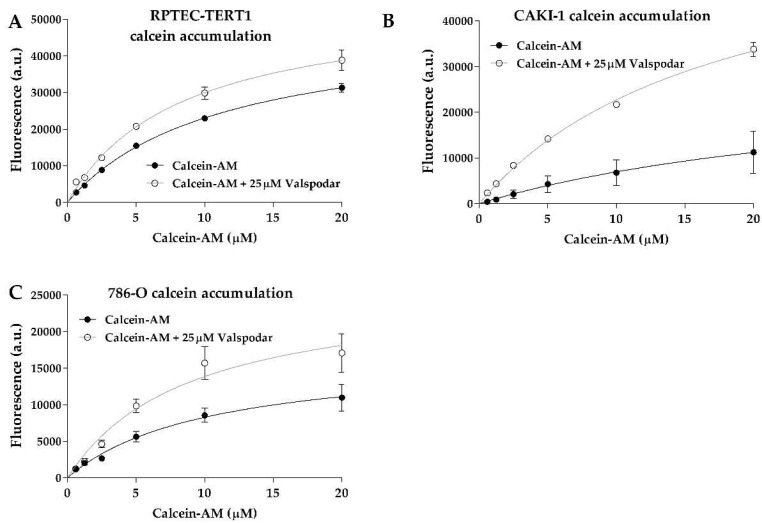
**P-glycoprotein drug transport activity.** Uptake of calcein-AM in the presence or absence of the P-gp inhibitor valspodar. Functional data derived from fluorescent assays were fitted according to Michaelis–Menten kinetics to evaluate the drug transport activity in RPTEC-TERT1 cells (**A**), CAKI-1 cells (**B**), and 786-O cells (**C**). An increase in calcein retention after inhibition indicated the presence of P-gp-mediated efflux activity. Functional data were obtained from a minimum of two independent assays, performed in triplicate.

**Figure 3 ijms-23-10177-f003:**
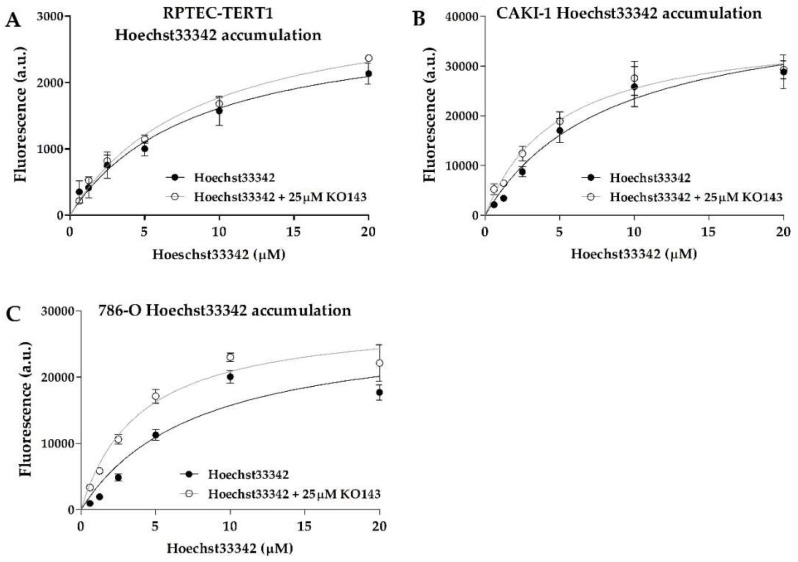
**Breast cancer resistance protein drug transport activity.** Uptake of Hoechst33342 in the presence or absence of the BCRP inhibitor KO143. Functional data derived from fluorescent assays were fitted according to Michaelis–Menten kinetics to evaluate the drug transport activity in RPTEC-TERT1 cells (**A**), CAKI-1 cells (**B**), and 786-O cells (**C**). An increase in Hoechst33342 retention after inhibition indicated the presence of BCRP-mediated efflux activity. Functional data were obtained from a minimum of two independent assays, performed in triplicate.

**Figure 4 ijms-23-10177-f004:**
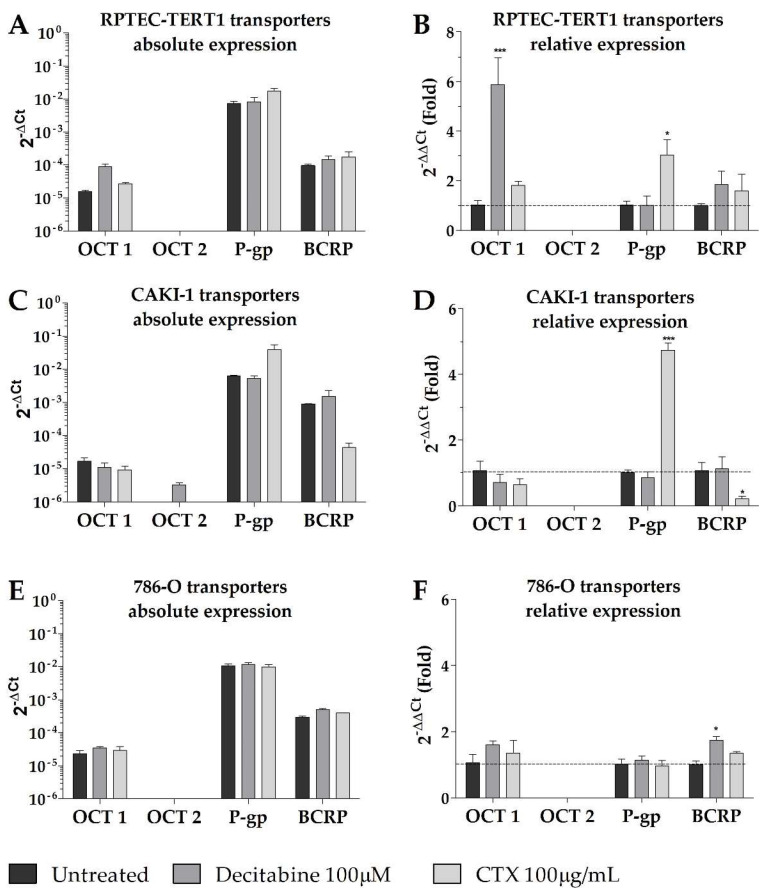
**Gene expression of key renal drug transporters in non-malignant and malignant RPTECs after treatment with methylation or EGFR inhibitors.** Expression is depicted as absolute, with 2^−ΔCt^ values expressed in a log10 scale (**A**,**C**,**E**) and as relative, with 2^−ΔΔCt^ values (fold) depicted in a linear scale (**B**,**D**,**F**). OCT2 was not found to be expressed in any of the cell lines tested and its expression was recovered in CAKI-1 cells (**C**) after decitabine exposure. Results represent data collected from three independent experiments. (* *p* < 0.05, *** *p* < 0.01).

**Figure 5 ijms-23-10177-f005:**
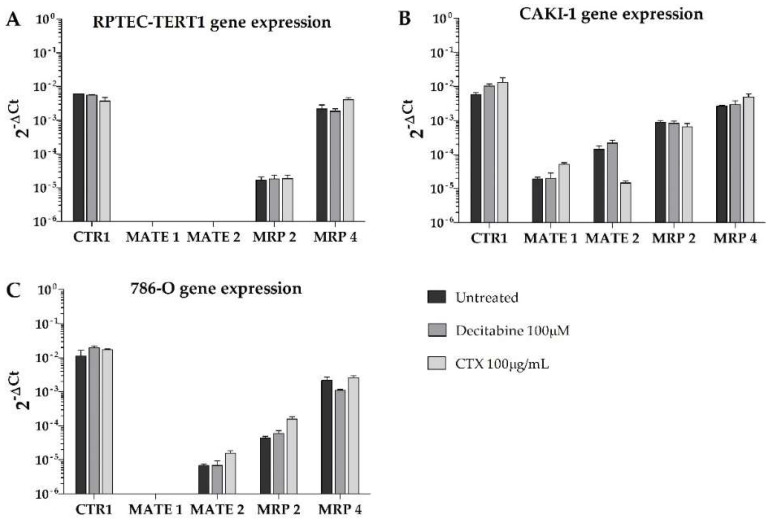
**Gene expression of relevant drug transporters in the kidney after treatment with methylation or EGFR inhibitors**. Expression is depicted as absolute, with 2^−ΔCt^ values expressed in a log10 scale. RPTEC-TERT1 cells (**A**) did not express MATE1 and MATE2 transporters; in CAKI-1 cells (**B**), MATE1 and MATE2 were sensitive to CTX exposure; and in 786-O cells (**C**), MATE1 was absent. Results represent data collected from three independent experiments.

**Figure 6 ijms-23-10177-f006:**
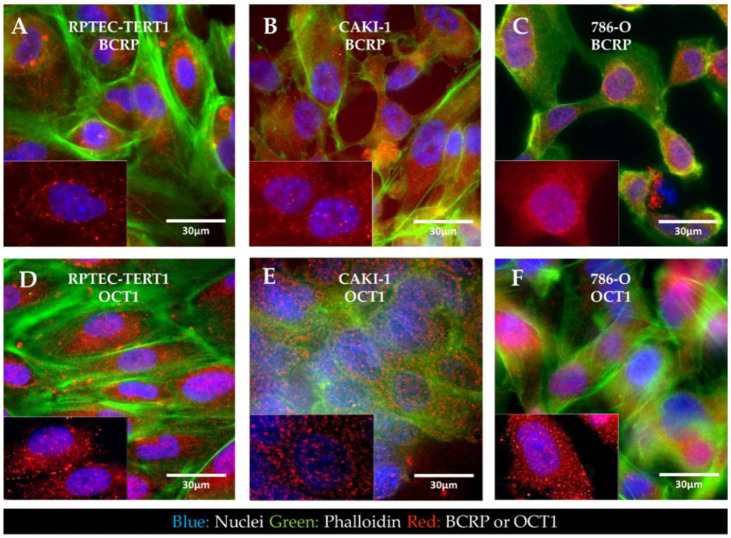
**Immunofluorescent localization of BCRP and OCT1 in non-tumor and RCC cell lines.** In RPTEC-TERT1 cells (**A**), CAKI-1 cells (**B**), and 786-O cells (**C**), BCRP expression was seemingly dispersed through the cytoplasm and in the vicinity of the nuclear region, with no predominant localization observed in the membrane. Similar to BCRP, OCT1 expression in RPTEC-TERT1 cells (**D**), CAKI-1 cells (**E**), and 786-O cells (**F**) was dispersed in the cells. All panels represent a composite image, including a nuclear tracer (Hoeschst33342, blue), f-actin (Alexa488, green), and BCRP or OCT1 (Alexa555, red). Amplified sections in the right bottom corner of the panels represent the nuclear and respective transporter staining. All images presented were acquired with a magnification of 60×. The bottom-left panel in each image represents the amplified detail of the respective image without the phalloidin staining.

**Figure 7 ijms-23-10177-f007:**
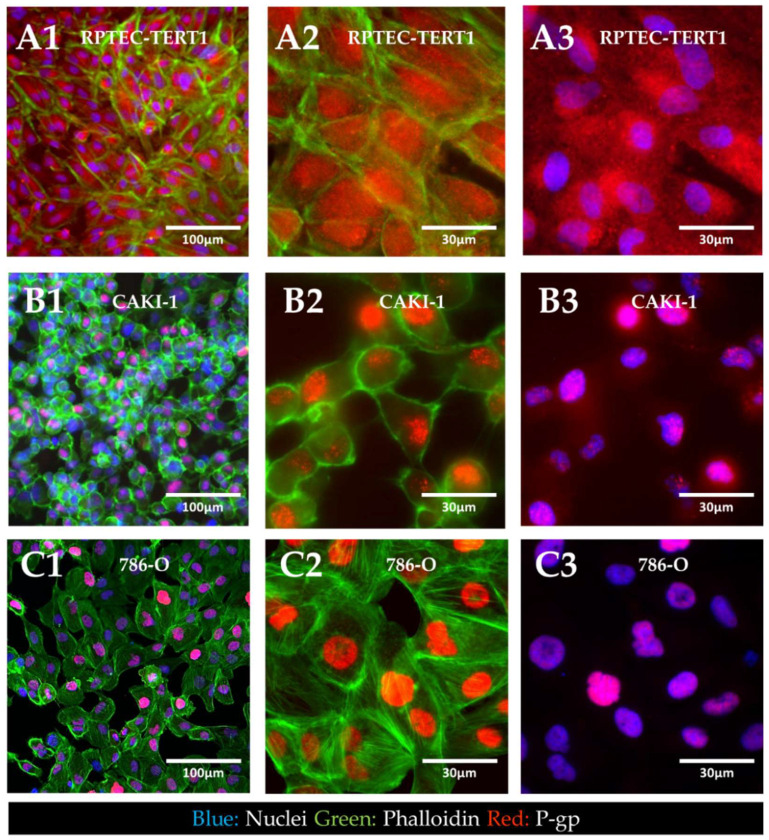
**Immunofluorescent localization of P-gp in non-tumor and RCC cell lines.** In RPTEC-TERT1 cells (**A**), P-gp expression was seen dispersed throughout the cytoplasm, with no predominant localization in particular cellular structures. In CAKI-1 (**B**) and 786-O (**C**) cells, P-gp expression was seemingly confined to the nuclear region of the cells, which was evident from the co-localization of the nuclear tracer (blue) and the P-gp stain (red). Panels **A1**, **B1**, and **C1** represent a composite image, including a nuclear tracer (Hoechst33342, blue), f-actin (Alexa488, green), and P-gp (Alexa555, red). Panels **A2**, **B2**, and **C2** represent a composite image of P-gp and f-actin. Panels **A3**, **B3**, and **C3** represent a composite image of P-gp and the nuclear staining. The images in panels **A1**, **B1**, and **C1** were acquired with a magnification of 20× to provide an overview of cell density and monolayer growth. The images in panels **A2,A3**, **B2,B3**, and **C2,C3** were acquired with a magnification of 60× to capture the cellular morphology in detail.

**Figure 8 ijms-23-10177-f008:**
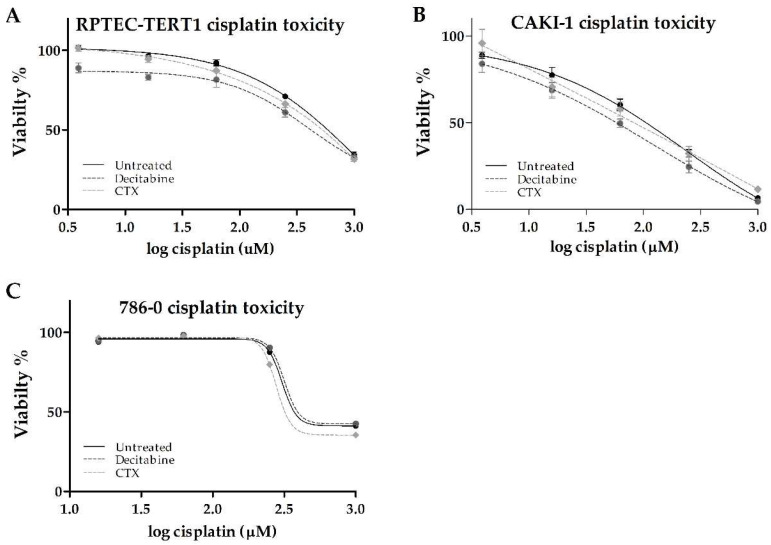
**Effects of methylation and EGFR inhibition on the toxicity of cisplatin**. In RPTEC-TERT1 (**A**) and CAKI-1 (**B**) cells, decitabine enhanced toxicity, while in 786-O cells (**C**), no effect was observed.

**Figure 9 ijms-23-10177-f009:**
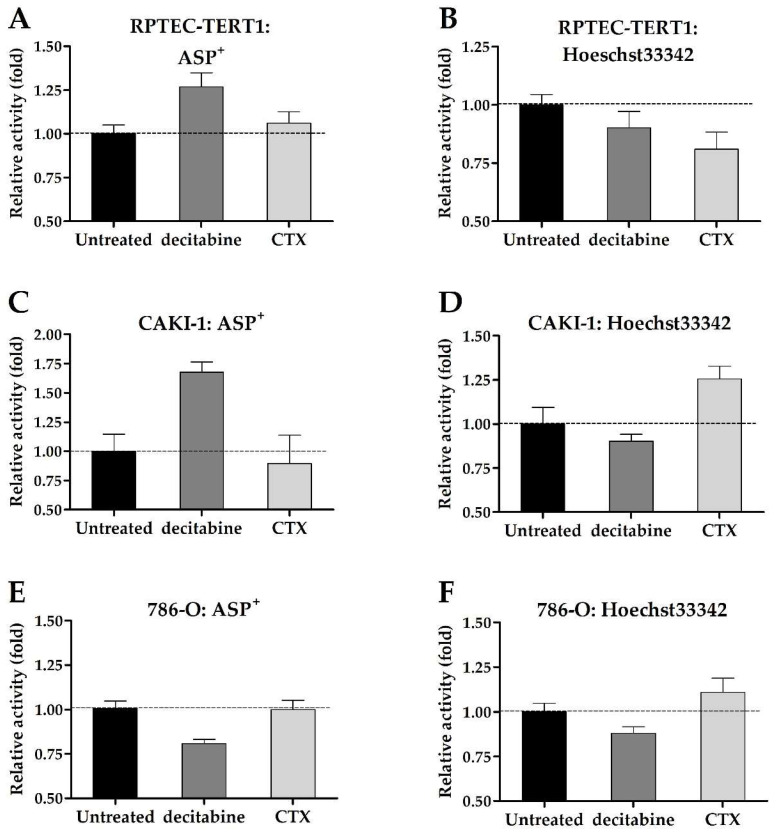
**Accumulation of ASP^+^ and Hoecht33342 after decitabine or CTX treatment.** In RPTEC-TERT1 cells, ASP^+^ (**A**) uptake was upregulated after decitabine treatment and Hoechst33342 (**B**) retention was reduced after CTX exposure. In CAKI-1 cells, decitabine yielded an increased ASP^+^ uptake (**C**) while CTX increased Hoechst33342 retention (**D**). In 786-O cells, decitabine reduced ASP^+^ uptake (**E**) and Hoechst33342 retention (**F**). The accumulation of fluorescent substrates after treatment with decitabine or CTX was low as opposed to the untreated samples. Therefore, the relative activity scale (y-axis) was set at 0.5-fold to demonstrate the differences. All assays represent a minimum of three independent assays performed in triplicate.

**Figure 10 ijms-23-10177-f010:**
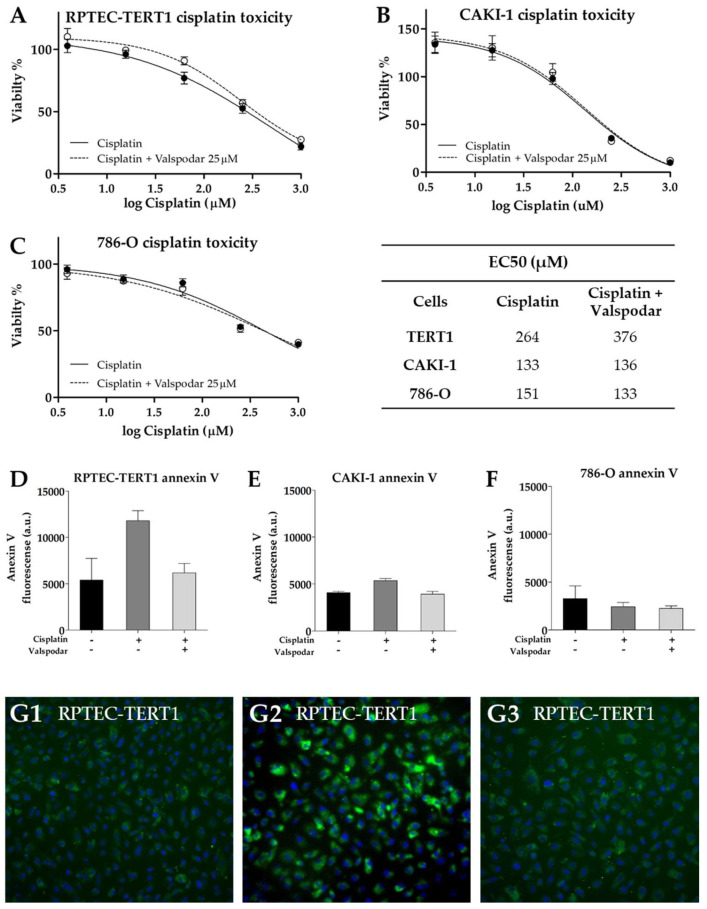
**Effects of P-gp inhibition on cisplatin toxicity.** The inhibitor valspodar reduced the cisplatin toxicity in RPTEC-TERT1 cells (**A**), but had no effect on CAKI-1 (**B**) and 786-O (**C**) cells. Similar effects were observed with the fluorescently labeled apoptosis marker annexin V, after a 200 µM cisplatin exposure. Annexin V fluorescence increased in RPTEC-TERT1 cells after cisplatin treatment (**D**,**G2**) relative to the untreated cells (**G1**), an effect that was absent when cisplatin was co-exposed with the P-gp inhibitor valspodar (**G3**). Annexin V fluorescence was not affected by cisplatin in CAKI-1 (**E**) and 786-O (**F**) cells. The concentration of 200 µM was selected in order to induce limited toxicity and prevent excessive cell death. Images in panels G1–3 were acquired with a magnification of 20×.

**Table 1 ijms-23-10177-t001:** Apparent K_m_ and V_max_ parameters derived from fluorescent functional assays in RPTEC-TERT1, CAKI-1, and 786-O cells.

Cells	ASP^+^				Calcein				Hoechst33342			
	K_m_ (µM)		V_max_ (a.u.)		K_m_ (µM)		V_max_ (a.u.)		K_m_ (µM)		V_max_ (a.u.)	
	−Inh	+Inh	−Inh	+Inh	−Inh	−Inh	−Inh	+Inh	−Inh	+Inh	−Inh	+Inh
**TERT1**	-	-	-	-	11.2	11.2	48,779	54,835	8.4	8.7	2975	3311
**CAKI-1**	32.8	74.7	5385	7443	31.4	31.4	28,813	63,237	3.8	3.1	1746	2224
**786-O**	-	-	-	-	10.3	10.3	16,798	26,124	7.8	3.9	27,961	29,092

**Table 2 ijms-23-10177-t002:** Relative gene expression (2^−ΔΔCt^; fold) of the transporter genes analyzed relative to the untreated control. Values highlighted in green represent an upregulation in gene expression and values highlighted in red represent a downregulation in gene expression.

Transporter	TERT1		CAKI-1		786-O	
	Decitabine	CTX	Decitabine	CTX	Decitabine	CTX
**OCT1**	5.9 ± 1.5	1.8 ± 0.2	0.7 ± 0.4	0.6 ± 0.3	1.6 ± 0.2	1.3 ± 0.5
**OCT2**	-	-	-	-	-	-
**P-gp**	1.0 ± 0.5	3.0 ± 0.9	0.8 ± 0.2	4.7 ± 0.3	1.1 ± 0.2	0.9 ± 0.3
**BCRP**	1.8 ± 0.7	1.6 ± 0.9	1.1 ± 0.3	0.2 ± 0.09	1.7 ± 0.2	1.7 ± 0.07
**CTR1**	0.9 ± 0.06	0.9 ± 0.1	1.8 ± 0.3	3.1 ± 0.3	1.2 ± 0.2	1.0 ± 0.06
**MATE1**	-	-	1.0 ± 0.6	2.7 ± 0.4	-	-
**MATE2**	-	-	1.7 ± 0.5	0.1 ± 0.02	1.0 ± 0.6	2.3 ± 0.6
**MRP2**	1.17 ± 0.5	1.2 ±0.5	0.9 ± 0.2	0.8 ± 0.2	1.3 ± 0.4	3.5 ± 0.9
**MRP4**	1.0 ± 0.3	2.0 ± 0.5	1.1 ± 0.5	2.3 ± 0.1	0.5 ± 0.05	1.2 ± 0.2

**Table 3 ijms-23-10177-t003:** Cisplatin EC50 values (µM) after pre-treatment with decitabine or CTX.

Cells	Untreated	CTX	Decitabine
**TERT1**	1101 ± 81.6	-	397.9 ± 27.9
**CAKI-1**	236.4 ± 32.6	-	123.4 ± 39.5
**786-O**	305.5 ± 44.7	279.0 ± 9.2	317.2 ± 91.9

## Data Availability

All data collected during the execution of the present study can be made available by the corresponding author upon request.

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
