# Peer review of "In Vitro Characterization of Renal Drug Transporter Activity in Kidney Cancer"

_ijms, 2022, doi:10.3390/ijms231710177_

Round 1
Reviewer 1 Report
please find attached.

Author Response
Reply to reviewer
The authors are much appreciated for the constructive and positive appraisal of our manuscript entitled: ´´The characterization of renal drug transporters activity in kidney cancer in vitro´´. We have taken all remarks into consideration and improved our manuscript accordingly. Special emphasis has been given to the figure presentation and proofreading of the revised version. Please find below point-by-point answers to all comments.
ijms-1884762
Title: “The characterization of renal drug transporters activity in kidney cancer in vitro”
Authors: Caetano-Pinto et al.
Here, the authors analyzed the activity, expression, and potential regulation of renal drug transporters in renal cell carcinoma, as well as non-malignant renal cells. First, uptake and efflux activity of three cell lines was assessed. Secondly, they checked alterations of transporter expression after methylation or EGFR inhibition, and their potential effects on cisplatin cytotoxicity. Additionally, cellular localization of analyzed transporters was determined by IF-staining.
Clearly, the authors’ work is appreciated, and the study is relevant for the kidney cancer field. However, I feel that the authors need to significantly improve the manuscript presentation. In summary, I am unfortunately unable not recommend acceptance of the report for publication in ijms in its current form.
I have listed some points that the authors should address before publication. I would also highly recommend a professional proofreading service!
- Presentation of figures must be improved: resolution of transport curves seems low, and pattern in the background should be removed
RE: All figures including graphics and immunofluorescence images have been revised to improve resolution and eliminate any background effect. We reiterate that in addition to the manuscript submission file, a separate folder including high-definition images of all figures was also provided.
- Authors should comment on how/why the analyzed drug transporters were chosen. “Key renal drug transporters” were analyzed, but what is about the others, OATs, MATE and so on…?
RE: We very much appreciate this remark and acknowledge that referring to a selected number of drug transporters as ´´key´´ is subjective, in particular when considering renal transporters. The overall list of renal drug transporters analyzed at the gene expression level in our study (except CTR1 and OCT1) are recommended for pre-clinical evaluation in the guidelines for ´´In Vitro Metabolism and Transporter Mediated Drug-Drug Interaction Studies´´ of Federal Drug Administration – FDA. This recommendation reflects their physiological impact on renal physiology, including the extensive array of substrates they secrete and proven impact on nephrotoxicity. From these transporters, P-gp and BCRP were included considering that they have the most known renal substrates and contribute extensively to RPTEC efflux (https://transportal.compbio.ucsf.edu/). OAT1 and OAT3 are not expressed across all cell lines analyzed, hence could not be characterized. OCT2 was chosen to represent uptake drug transport activity considering its widely reported expression in human kidney cortex. However, due to its limited expression in the cell lines used in the study, OCT1 was also selected to evaluate cation uptake activity. Moreover, the fluorescent functional assays used were also developed based on the activity of P-gp, BCRP and OCTs. In the revised version of our manuscript the wording ´´key renal drug transporters´´ has been changed to minimize the subjectivity of the transporters selected and the mention of the FDA guidelines is reiterated. Further references (32,33,34) were added to our revised manuscript to support the selection of the renal drug transporters in question.
- English needs (professional) proofreading! A lot of spelling and punctuation errors! For example, one sentence (on p. 7, l. 212 f.) contains 8 (!) errors: Moreover, the phallodin satining reveals that RPTEC-TERT1 cells maintains a more charateristic epthitilial morphology, with f-actin filamnets predominantly concentrating in the cellular boundries and cells growing into a regular monolayer, contrary to the tumor cell lines.
RE: We are thankful to the reviewer for pointing out these mistakes. We have thoroughly checked our manuscript for typos and grammatical misconstructions. The revised manuscript submitted has been professionally proofread following all recommended changes.
- Title might be changed to “In vitro characterization of renal drug transporter activity in kidney cancer cells”
RE: We appreciate this suggestion and have changed the title of the revised manuscript, which better reflects the in vitro nature of our work.
- Numbering/appearance of figures is confusing, why referring to Figure 7 in previous section (after Figure 4). Same for Table 2. Figures/Tables should be referred to in order of appearance. Please revise and re-arrange figures/Tables accordingly.
RE: The figure and table numberings have been harmonized in the revised version of our manuscript to avoid any confusion and facilitate readability. Figure 7 and Table 2 in our original submission have been repositioned after Figure 4, so the Figures and Tables relating to the gene expression analysis are contiguous (Figures 4 and 5, and Table 2 in our revised manuscript).
- Figure legends also confusing, eg Figure 5e: red staining – figure says it is BCRP, but in legend its OCT1?! Please revise!
RE: The figure legends and panels referring to the immunofluorescent staining (Figures 6 and 7 in the revised manuscript) have been re-arranged to improve the data presentation. The mistakes in Figure 5 (revised: Figure 6) have been corrected.

Reviewer 2 Report
The authors present an interesting study examining the differential profiles of drug transporters across a number of kidney cell lines. Specifically, the authors question the validity of certain cell lines and types in the context of cancer research owing to the expression profiles of particular drug transporters in control conditions, and following treatment with specific therapeutics. In reviewing the gene expression, protein localisation, and viability of the different cell types to these treatments, the reviewers conclude that there are particular limitations to certain cells and cell lines in utilising them for renal research, while highlighting the distinct advantages and disadvantages of each. Taken together, research such as this is critically important to the field, and I commend the authors for pursuing such.
In reviewing the manuscript, I did note a number of points for the authors attention. The following should be considered when preparing a suitable revision.
1. The writing in the manuscript is good for the most part, however there are typographical errors within, in both the text and the figures e.g., Figure 1C. The authors should review the manuscript for these instances and amend where incurred.
2. In the methods, it would be preferable if the authors included information on where the reagents/consumables were bought, with details were pertinent on the specific product codes.
3. Were the primers/probes used in this study MIQE guidelines compliant?
4. For the expression of targets, in particular Oct2, was any control sample included to validate the protocol was working? That this protocol could indeed efficiently detect the levels of Oct2?
5. Why did the authors opt for absolute values for the gene expression instead of examining relative values?
6. The authors explore gene expression and also protein localisation, but do not perform any protein work (i.e. Western blots) to examine protein expression levels of transporters of interest. Why was this not pursued?
7. While not essential, it might be advisable to adjust the scale bars such that they are more relative to one another. At the moment the differences within each graph are clear, however in comparing the different cell types there are stark differences and the scale is somewhat masking this.
8. In examining the annexin V images and fluorescence readings, the change in level of fluorescence in the images appears to be much stronger than that indicated by the plate reader measurements. The authors must clarify on the differences between these data.
9. In table 2, it should be indicated clearly as to what the green and red shading alludes to.
10. In the microscopy images, a legend should be included for what each colour represents. At the moment, the legend that is affixed to each image is difficult to read, and it might be worthwhile formatting a new clearer legend, or else referring to it in the figure legend, as to what each colour represents.
11. Magnification and scale cars should be included in the microscopy images/legends.
Author Response
Reply to reviewer
The authors are much appreciated for the constructive and positive appraisal of our manuscript entitled: ´´The characterization of renal drug transporters activity in kidney cancer in vitro´´. We have taken all remarks in consideration and improved our manuscript accordingly. Please find below a point-by-point answers to all comments.
Comments and Suggestions for Authors
The authors present an interesting study examining the differential profiles of drug transporters across a number of kidney cell lines. Specifically, the authors question the validity of certain cell lines and types in the context of cancer research owing to the expression profiles of particular drug transporters in control conditions, and following treatment with specific therapeutics. In reviewing the gene expression, protein localisation, and viability of the different cell types to these treatments, the reviewers conclude that there are particular limitations to certain cells and cell lines in utilising them for renal research, while highlighting the distinct advantages and disadvantages of each. Taken together, research such as this is critically important to the field, and I commend the authors for pursuing such.
In reviewing the manuscript, I did note a number of points for the authors attention. The following should be considered when preparing a suitable revision.
- The writing in the manuscript is good for the most part, however there are typographical errors within, in both the text and the figures e.g., Figure 1C. The authors should review the manuscript for these instances and amend where incurred.
RE: We acknowledge that the proofing of our manuscript requires improvement and the revised version has been thoroughly checked for typos and grammatical misconstructions.
- In the methods, it would be preferable if the authors included information on where the reagents/consumables were bought, with details were pertinent on the specific product codes.
RE: All missing supplier references for reagents, chemicals, and consumables have been added to our revised manuscript, in particular in section 4. A list of reagents is also now include in Appendix B TableB2. We also reiterate that the information regarding primers and antibodies used is listed in the Appendix B as well.
- Were the primers/probes used in this study MIQE guidelines compliant?
RE: The TaqMan probes used in our study are compliant with MIQE guidelines (https://www.thermofisher.com/de/de/home/life-science/pcr/real-time-pcr/real-time-pcr assays/why-choose-taqman-assays/publish-real-time-pcr-results.html).
- For the expression of targets, in particular Oct2, was any control sample included to validate the protocol was working? That this protocol could indeed efficiently detect the levels of Oct2?
RE: We acknowledge the reviewer's concerns regarding the validation of experimental protocols used. We have previously validated OCT2, and the other drug transporter probes used, by analyzing the gene expression of the targets in human primary RPTEC (data included in appendix) and in a human renal cortex tissue sample (data pending publication). Moreover, the OCT2 antibody used to determine the expression and localization of OCT2 was also validated in human primary RPTEC and a HEK-OCT2 overexpressing cell lines (data pending publication).
- Why did the authors opt for absolute values for the gene expression instead of examining relative values?
RE: In our manuscript, we present both absolute and relative gene expression values. We opted to include absolute expression values since they inform if the expression of a particular target is high or low relative to the housekeeping control. In the case of Oct2, only CAKI-1 cells expressed this transporter after treatment with Decitabine, albeit to a less extent. This information could be better presented in terms of absolute expression, given that Oct2 could not be normalized and expressed as relative.
- The authors explore gene expression and also protein localisation, but do not perform any protein work (i.e. Western blots) to examine protein expression levels of transporters of interest. Why was this not pursued?
RE: We appreciate this very pertinent remark and acknowledge that determining protein expression could complement our results, in addition to the gene expression and localization of drug transporters. For the scope of our present manuscript, which is mostly focused on the function of renal drug transporters, we choose gene expression to indicate the presence or absence of particular drug transporters and as an indication of their relative expression levels. Immunofluorescent localization was used since the cellular distribution of drug transporters substantially impacts their activity. Renal drug transport in health and disease is an active research topic in our group and ongoing research is focused on investigating the regulatory mechanisms of drug transport in kidney cancer. For this purpose, we are employing global-proteomics and, prospectively, western-blotting, to look further into transporter expression and associated proteins in both RCC cell lines and RCC human samples. In itself we believe that this work warrants an original publication when completed.
- While not essential, it might be advisable to adjust the scale bars such that they are more relative to one another. At the moment the differences within each graph are clear, however in comparing the different cell types there are stark differences and the scale is somewhat masking this.
RE: In the revised version of our manuscript the scale bars in the gene expression Figures have been harmonized to facilitate comparisons between the different cell lines.
- In examining the annexin V images and fluorescence readings, the change in level of fluorescence in the images appears to be much stronger than that indicated by the plate reader measurements. The authors must clarify on the differences between these data.
RE: We thank the reviewer for pointing out this discrepancy. The annexin-V-FITC fluorescence level is likely on the lower detection limit of the microplate reader used, therefore retrieving a high background. The microplate reader measurements shown represent the absolute fluorescence values where the background was not excluded (the reference background used was the annexin-V incubation buffer). The fluorescence-microscope used to acquire the annexin-V-FITC images could capture bigger differences between the different conditions given its sensitivity. In our revised manuscript we have excluded the background values from the microplate measurements.
- In table 2, it should be indicated clearly as to what the green and red shading alludes to.
RE: Table 2 now clearly indicates the significance of the green and red shadings used, which illustrate and up-and down-regulation of expression, respectively.
- In the microscopy images, a legend should be included for what each colour represents. At the moment, the legend that is affixed to each image is difficult to read, and it might be worthwhile formatting a new clearer legend, or else referring to it in the figure legend, as to what each colour represents.
RE: The microscope images in our revised manuscript have been revised to improve readability. The affixed legends have been replaced by headings in the figure panels and the color coding is now included in a more visible legend under the main figure panels. Moreover, Figures 5 and 6 have been repositioned as Figures 6 and 7 in our revised manuscript.
- Magnification and scale cars should be included in the microscopy images/legends.
RE: We acknowledge this missing information and our revised images now include scale bars (Figures 6 and 7 of the revised manuscript) and the respective image magnifications used are included in the figure legends.

Round 2
Reviewer 1 Report
The authors provided an improved version of the manuscript which is acceptable for publication in IJMS. However, there are still spelling errors which have to be removed, eg. "sataining", and "Cisplain" in a figure table.
Reviewer 2 Report
The authors have responded positively to my comments, and for that the manuscript is much improved.